

# Association between *ACE* and *ACTN3* genes polymorphisms and athletic performance in elite and sub-elite Chinese youth male football players

Shidong Yang[1,2], Wentao Lin[3], Mengmeng Jia[2] and Haichun Chen[2]

[1] Department of Physical Education, Nanjing Xiaozhuang University, Nan Jing, China
[2] Department of Physical Education and Sports Science, Fujian Normal University, Fu Zhou, China
[3] Department of Physical Education, Zhuhai University of Science and Technology, Zhuhai, China

## ABSTRACT

**Background.** Previous studies have shown controversial relationships between *ACE* I/D and *ACTN3* R577x polymorphisms and athletic performance. Therefore, the aim of this study was to assess athletic performance indicators of Chinese youth male football players with different ACE and ACTN3 gene profiles.

**Methods and Materials.** This study recruited 73 elite (26 13-year-olds, 28 14-year-olds, and 19 15-year-olds) and 69 sub-elite (37 13-year-olds, 19 14-year-olds, and 13 15-year-olds) and 107 controls (63 13-year-olds, and 44 14-year olds aged 13–15 years, all participants were of Chinese Han origin. We measured height, body mass, thigh circumference, speed, explosive power, repeat sprints ability, and aerobic endurance in elite and sub-elite players. We used single nucleotide polymorphism technology to detect controls elite and sub-elite players' *ACE* and *ACTN3* genotypes, Chi-squared ($\chi^2$) tests were employed to test for Hardy-Weinberg equilibrium. $\chi^2$ tests were also used to observe the association between the genotype distribution and allele frequencies between controls and elite and sub-elite players. The differences in parameters between the groups were analyzed using one-way analysis of variance and a Bonferroni's *post-hoc* test, with statistical significance set at $p \leq 0.05$.

**Results.** (1) The genotype distribution of the *ACE* I/D and *ACTN3* R577x polymorphisms in controls, elite and sub-elite football players were consistent with Hardy-Weinberg equilibrium, except for the *ACE* genotype distribution of sub-elite players. (2) The RR and DD genotypes were significantly different between elite and sub-elite players ($p = 0.024$ and $p = 0.02$, respectively). (3) Elite players were more likely to have the RR genotype and less likely to have the DD genotype compared with sub-elite players. (4) Both elite and sub-elite RR players' Yo-yo intermittent recovery level 1 (YYIR1) running distance was significantly longer than that of RX players ($p = 0.05$ and $p = 0.025$, respectively). However, there was no significantly different in YYIR1 running distance between elite and sub-elite RR players. (5) Elite XX players' VO$_2$ max was significantly higher than that of RX and sub-elite players.

**Conclusion.** These results indicate that *ACE* I/D and *ACTN3* R577x polymorphisms are not associated with muscle power in Chinese elite and sub-elite players. The XX genotype of ACTN3 is associated with the aerobic endurance of elite players.

Corresponding author
Wentao Lin, wentaoltg@163.com

# INTRODUCTION

Improving the athletic performance of elite athletes has always been an active research topic in the field of sports science. Although well-designed scientific training program and an athlete's hard work can improve athletic performance, some people can still perform a high level of athletic performance without high-intensity training (*Tucker & Collins, 2012*). It is well known that genetic factors have a great influence on athlete's strength, power, speed, endurance, psychological traits, and other phenotypes (*Ahmetov & Fedotovskaya, 2015*). An increasing number of studies have shown that there was a significant association between genes and athletic performance (*McAuley et al., 2021a*).

The angiotensin converting enzyme (*ACE*) and alpha-actinin-3 (*ACTN3*) polymorphisms have been intensively studied in athletic performance in endurance and strength/power-oriented sports events, and has been proven to be significantly associated with athletic performance (*Jeremic et al., 2019*).

$\alpha$-Actinin is an actin-binding protein, which is distributed in the Z line of skeletal muscle and combines with thin myofilament to maintain the orderly arrangement and contractile function of muscle fibers. *ACTN3* is found only in fast-muscle fibers, which produce the contractions required for explosive force (*MacArthur & North, 2004*).The studies on the *ACTN3* gene have focused on the R577X polymorphism in exon 16, which causes the codon encoding the amino acid at position 577 to change from CGA (encoding arginine R) to TGA (termination signal, noncoding protein), resulting in the loss of *ACTN,* when the coding is terminated, $\alpha$-actinin-3 is deleted (*North et al., 1999*; *Malyarchuk, Derenko & Denisova, 2018*; *Vincent et al., 2010*). *Yang et al. (2003)* studied Australian elite athletes and suggested that *ACTN* 3 was required for optimal fast-muscle fiber performance in strength and speed athletes, and that lack of ACTN3 may be beneficial for endurance athletes.

Many studies have reported that the proportion of RR genotype of *ACTN3* in elite power-oriented sports (*e.g.*, rowing, 100-m sprint, long jumpers, and weight lifting) is significantly higher than XX genotypes across cohorts in United States, Poland, Italy, Japan, China, and Russia (*Ahmetov et al., 2011*; *Chiu et al., 2011*; *Jastrzebski et al., 2014*; *Druzhevskaya et al., 2008*; *Mikami et al., 2014*; *Yang et al., 2017*). *McAuley et al. (2021b)* conducted a meta-analysis on 17 studies involving football players and discovered that the R allele was significantly associated with professional football players, and that the RR genotype may be more associated with Caucasian players, while the RR genotype may be more associated with Brazilian players.

Many authors have suggested that the *ACTN3* RR genotype was associated with the power-orientated phenotypes of football players (*Atabaş et al., 2020*; *Massidda et al., 2012a*; *Massidda et al., 2012b*; *Pimenta et al., 2013*; *Dionísio et al., 2017*). *Pimenta et al. (2013)* suggested that Brazilian professional players with *ACTN3* RR were faster and jumped higher compared to XX players, and that XX players had higher aerobic capacity than RR players. However, *Coelho et al. (2016)* posited that the *ACTN3* genotype in Brazilian

players was not associated to physical performance. This discordance may be related to the fact that Brazil is a country with multiple ethnicities, and that the proportion of ethnic groups in each region is different (*McAuley et al., 2021b*).

The *Actn3* $^{-/-}$ knockout mouse model was applied to explain the mechanism by which $\alpha$-actin-3 alters skeletal muscle function (*MacArthur et al., 2007*). Compared with wild-type mice, ACTN3 knockout mice exhibited decreased fast-twitch fiber diameter and decreased muscle mass; their grip strength was significantly reduced; and their endurance was significantly augmented (*MacArthur et al., 2008*). Anatomical analysis revealed that the activities of anaerobic metabolic enzymes decreased and the activities of aerobic metabolic enzymes increased in fast-twitch muscle fibers of knockout mice, but that the distribution of muscle fiber types did not change significantly (*MacArthur et al., 2008*). *Vincent et al. (2007)* conducted muscle biopsies of people with different *ACTN3* genotypes and found that the area and number of fast-twitch fibers in individuals with the RR genotype were significantly greater than those with the XX genotype. These authors also postulated that the effects of $\alpha$-actin-3 deficiency on explosive power were related to the proportion of muscle fiber types due to a reduction in fast-twitch fibers (*Vincent et al., 2007*).

Studies have shown that the XX genotype was associated with VO$_2$ max (*Silva et al., 2015*; *Pimenta et al., 2013*). *Shephard (2015)* found in a study of the effect of *ACTN3* on blood iron metabolism that the reduction in iron could only be observed in the RR/RX genotype, and that there was a lower proportion of hematuria in long-distance runners with the XX genotype. *Grygorczyk & Orlov (2017)* suggested that the erythropoietin levels were elevated in the XX genotype, while the RR genotype reflected a higher proportion of hemolytic metabolites and myoglobinuria impairment. *Sierra et al. (2019)* demonstrated that endurance athletes with the RR genotype possessed higher proportion of hematuria, leukocyturia, iron deficiency, creatinine, myoglobin, and bilirubin imbalances than the XX genotype. RR-genotype endurance athletes also still maintained low levels of red blood cells and iron 15 days after a race, while XX genotype athletes were more likely to maintain higher levels of red blood cells and iron.

*ACE* is another commonly investigated gene that affects athletic performance (*McAuley et al., 2021a*). The angiotensin I converting enzyme catalyzes the degradation of the inactive decapeptide angiotensin I and subsequently generates the physiologically active peptide angiotensin II, an oligopeptide that consists of eight amino acids and affects several systems by binding to specific receptors in the body (*Dzau, 1988*; *Munzenmaier & Greene, 1996*). *ACE* is a key enzyme in the renin-angiotensin system, which is widely distributed in various humantissues, including skeletal muscle, and it plays a biological role in degrading bradykinin and converting angiotensin I to angiotensin II (*Woods, 2009*; *Jones & Woods, 2003*; *Erdös & Skidgel, 1987*). Activation of the renin-angiotensin system has been associated with mechanical, metabolic, and biochemical changes in skeletal muscle (*Schaufelberger et al., 1996*).

Many previous studies have suggested that the II genotype and I allele were significantly associated with elite endurance athletes (*Ahmetov & Fedotovskaya, 2015*; *Woods, 2009*; *Alvarez et al., 2000*; *Gayagay et al., 1998*; *Scanavini et al., 2002*), and that the D allele was significantly associated with muscle-power athletes (*McAuley et al., 2021a*). This is because

the I allele may reduce the activity of the ACE enzyme (*Ma et al., 2013*; *Pescatello et al., 2019*), thereby improving endurance performance in humans, while the D allele is associated with increased baseline muscle volume and the proportion of fast contraction muscle fibers with greater power performance (*Eider et al., 2013*).

*ACE* promotes the conversion of angiotensin I (Ang I) to Ang II, and Ang II binds to receptors, causing vasoconstriction and regulating electrolyte balance. *ACE* is abundantly present on the membrane surface of vascular endothelial cells so that endogenous Ang I and bradykinin can be transformed by *ACE*, and A*CE* thus determines Ang II levels. Although *ACE* levels in plasma are less affected by environmental, humoral, and metabolic factors, individual differences are large (*Gao, Chen & Dong, 2006*). *Vaughan et al. (2013)* hypothesized that the presence of the I allele led to a diminution in ACE activity in serum, an attenuation in ACE gene transcription and expression, and a decrease in Ang II production capability. *Danser et al. (2007)* suggested that subjects with the D allele of *ACE* had higher serum and tissue *ACE* activity, resulting in a greater conversion of angiotensin I to angiotensin II. *Williams et al. (2000)* also argued that the II genotype could improve the mechanical efficiency of muscle work (which may be related to the increase in slow-twitch muscle fibers) because *ACE* enzyme activity associated with the II genotype is low, such that the local nitric oxide concentration in skeletal muscle increases and thereby enhances mitochondrial respiratory efficiency and skeletal muscle contractile function.

The ability to repeated high intensity exercise actions has a crucial effect on the performance of football players in a match (*Sarmento et al., 2018*; *Kelly & Williams, 2020*). It was reported that the D allele or R allele carriers had greater muscle strength and power and an increased percentage of fast-twitch muscle fibers (*Ahmetov et al., 2011*; *Erskine et al., 2014*; *Orysiak et al., 2014*; *Zhang et al., 2003*). However, not all  studies have confirmed these results (*Gentil et al., 2012*; *Garatachea et al., 2014*; *Ruiz et al., 2011*), so further research is needed to investigate the association between *ACE* and *ACTN3* genes and muscle phenotype. Previous studies were mainly designed simulate case-control evaluations of *ACTN3* and *ACE* polymorphism based on the exercise state, without investigating the characteristics of an athlete's competitive level. This study is the first to investigate the difference in the distribution of *ACE* and *ACTN3* polymorphism between elite and sub-elite football players, and the effects of genotypes (RR, RX, XX and II, ID, DD) on anthropometric and athletic performance of Chinese football players.

## METHODS
### Subjects
A total of 169  players were initially recruited from two football schools and one professional football club in China, and then 16 goalkeepers and 11 non-Han players were excluded. Thus, a total of 142 players of Chinese Han ethnicity (63 13-year-olds, 47 14-year-old-olds, and 32 15-year-olds) were ultimately included. We recruited 107 healthy, non-athletic controls, aged 13–14 years of junior high school boys. During the testing, subjects were permitted to withdraw in the case of injury or simply if they showed unwillingness to undertake the field test. Before the 2021–2022 seasons, the coaches defined players as

starting players and substitute players based upon training and competitive ability. This research thus designated 73 starting players as elite players and 69 substitutes as sub-elite players.

This study was conducted according to the Declaration of Helsinki and was approved by the Ethics Committee of Fujian Normal University. All players, coaches, and players' guardians were made aware of the test process and written informed consent was obtained.

### Experimental approach

Testing was divided into two days. On the morning of the first day, we measured the players' anthropometric indicators, including height, body mass and collected and stored the players' oral mucosa, and then tested their $20-m/30-m$ sprint, standing long jump, $2 \times 25$ m repeated sprint, and YYIR1. The next day, at the same time, we tested the players' 12-minute runs.

All players we recruited had no injuries and could withdraw from the test at the time of this study. DNA was extracted from the oral mucosa of every subject and was used to determine the subject's A*CE* and *ACTN3* genotype by PCR.

## Procedures
### Genotyping

DNA extraction: Mouth mucosa was collected *via* oral flocking swabs (Lang Fu Bio-instrument, Shanghai, China) and placed in a preservation solution.DNA was extracted from mouth mucosa using the TSINGKE silica gel adsorption kit (Qiagen Inc., Valencia, CA, USA).

### PCR amplification

The primer was synthesized by Qingke Biotechnology Co., and the amplification primer was also a sequencing primer (Table 1). A 50-µl reaction system was used for polymerase chain reaction (PCR) amplification (Table 2). The PCR conditions and procedure for *ACE* and *ACTN3* gene polymorphism analysis are shown in Table 3.

## Electrophoresis detection and sanger sequencing

The PCR products were subjected to agarose gel electrophoresis (2 µl sample + 6 µl bromophenol blue), the identification gel map was obtained at 300 V voltage for 12 min, and the amplification conditions were determined by the gel map.

PCR products were cut and recovered using DNA gel recovery kit; The BDT reaction system (normal 5 µl) was added in the order of 1 µl primer, 2 µl templates, and 1 µl BDT 1 µl betaine. After the completion of inspection, the membrane was covered, and units were balanced in a centrifuge and centrifuged at 4 °C and 4,000 RPM. We then covered the eight-tube caps, shook and mixed the reaction system, centrifuged it, and placed them in a PCR machine after completion to carry out the reaction. Ferrite beads (38 µl) were added to the centrifuged plate, and they were washed and purified with Magical Buffer. We copied the data from the sequencer and analyzed them with GeneMapper Analysis Software, version 4.1 (Applied Biosystems, Waltham, MA, USA).
**Table 1 Primers of *ACE* and *ACTN3* genes polymorphism for PCR.**

|  | Primer F | Primer R |
|---|---|---|
| ACE | TGGAGAGCCACTCCCATCCTTTC | TCCAGCCCTTAGCTCACCTCTGC |
| ACTN3 | GGGACACCAGCTGACACTTCCT | TGATGTAGGGATTGGTGGAGCA |

**Table 2 Components of the polymerase chain reaction PCR amplification system.**

| Components | Volume |
|---|---|
| Chingke Gold Mix | 47 ul |
| 10 $\mu$M Primer F | 1 ul |
| 10 $\mu$M Primer R | 1ul |
| Template (gDNA) | 1 ul |
| Total | 50 ul |

**Table 3 Polymerase chain reaction conditions for *ACE* and *ACTN3* genes.**

| Stage | Temperature | Time | Cycle number Number of cycles |
|---|---|---|---|
| Pre-denaturation | 98 °C | 2 min | 1 cycle |
| Cycle | 98 °C | 10 s | 30 cycles |
|  | TM °C | 10 s |  |
|  | 72 °C | 10 s |  |
| Extension | 72 °C | 5 min | 1 cycle |
| Preservation | 4 °C | – |  |

## Protocols

We measured height, body mass, and thigh circumference to evaluate the anthropometric characteristics of the subjects. We also determined sprinting, jumping, and aerobic performance to evaluate the physical characteristics of the subjects. All anthropometric measurements were conducted at 8:00 AM, and height and body mass were measured once, while the circumference of the thigh was measured twice. Our calculated intraclass correlation coefficients (ICC) were evaluated as "almost perfect" (ICC > 0.75) according to a previous study (*McGraw & Wong, 1996*).

The athletic performances were conducted following the guidelines of the China Football Association, which evaluate the physical abilities required by football players—including speed, jump length, and endurance. We measured 20-m and 30-m sprints, standing long jump (SLJ), 5× 25-m repeated sprints, Yo-yo intermittent recovery test level 1 (YYIR1), and a 12-minute running test. All subjects were requested not to exceed their normal training load two days before the test to exclude the effect of delayed muscle soreness on muscle function.

Participants sequentially performed 20-m and 30-m sprints, SLJ, 5 × 25-m RSA, and YYIR1. YYIR1 was conducted after completing the other tests; and the 12-minute run was evaluated at the same time on the next day. All physical measurements were executed twice with a rest period of three min or more between trials—except for the 5 × 25-m RSA,

YYIR1, and 12-minute running test that were each performed once. Before the test, players warmed up with 10 min of jogging and dynamic movements, and 5 min of sprinting and jumping with progressive intensity.

## Anthropometric measurements

Body sizes were measured with all football players' barefoot and wearing pants and a T-shirt. Height was measured with a YL–65S stadiometer (Yagami, Nagoya, Japan) to the nearest 0.1 cm, and body mass was measured with a XiaoMi body-fat monitor (XiaoMi, Beijing, China) to the nearest 0.1 kg.

The thigh circumference of the right thigh was measured with a standard tape measure to exactly 0.1 cm. We first measured the circumference of the thigh from a point five cm proximal to the upper end of the patella, and then measured the circumference of the thigh to obtain the average (*Mathur et al., 2008*). The intra-rater reliability ICC (3, 1) for thigh circumference was 0.9.

## Sprints measurement

Sprint times were assessed over 20 m and 30 m using timing gates (Brower Timing System, Draper, UT, USA). Subjects started 0.5 m behind the start line and ran maximally past the 30-m timing gate. Times were recorded to the nearest 0.01 s with the quicker of two attempts being used for the sprint score. The intra-rater reliability ICCs (3, 1) for the 20-m sprint and 30-m sprint tests were 0.9 and 0.89, respectively.

## Jump measurement

Standing long jump test was carried out to assess the players' explosive muscular power. All football players placed both feet behind a starting line and jumped as far as possible, while landing on both feet. The distance from the line to the player's closest heel was then measured with a measuring tape (*Glencross, 1966*; *Docherty, 1996*). The test was performed twice, and the longest jump distance between the two measurements was used for analysis. The intra-rater reliability ICC (3, 1) for the standing long jump tests was 0.86.

### *Repeat sprint ability measurement*

A 5 × 25-m repeated sprint was used to evaluate anaerobic endurance. On a 25-meter straight line, marker barrels were placed at intervals of 5 m. On hearing the command to "run", the participants ran from the starting line to the first marker barrel and knocked it down with their hands, then turned back to the starting line and again knocked down the marker barrel with their hands. This task was repeated for the second and remaining marker barrels. The runners sequentially knocked down all the marker barrels, and ultimately sprinted back to the starting line. Times were recorded to the nearest 0.01 s.

### *YoYo intermittent recovery level 1 measurement*

*Krustrup et al. (2003)* reported Yo-yo intermittent recovery level 1 (YYIR1) to be a valid and reliable test in assessing specific fitness for football. During the test, the players performed a series of 20-m runs—resting for 5 s every 40 m—and then ran with progressive increments in speed and for increasingly shorter time periods between changes. Failure to achieve the shuttle run in time on two occasions resulted in termination of the test. We therefore
recorded the distance when the players failed to cross the finish line on time for the second time.

### Aerobic fitness measurements

Cooper's 12-minute running test is a popular field test used to measure aerobic fitness, and it is also used to estimate an athlete's VO$_2$ max (*Conley et al., 1991*). The correlation of the 12-minute run test data with the laboratory-determined oxygen-consumption data was 0.897 (*Cooper, 1968*). The 12- minute run is one of the most commonly used methods of physical fitness training for Chinese youth football players. Prior to the test, the coach explained the test method and requirements to the players. The subjects were required to run as many laps as possible on a 400 m track for 12 min. After 12 min, the subjects were asked to stop running immediately. Then we recorded the subject's total distance (in meters) covered after 12 min. VO$_2$ max was predicted using the following formula:

$$VO_2 max(ml/kg/min) = (22.351 \times distance\ covered\ in\ kilometers) - 11.288.$$

### Statistical analysis

All statistical analyses were conducted using IBM-SPSS 26.0 for Windows. Chi-square ($\chi^2$) tests were employed to test for Hardy–Weinberg equilibrium (*Shenoy et al., 2010*). Fisher's exact tests were also employed to test genotype distribution and allele frequencies between the controls, elite and sub-elite players under the significant level at $p \leq 0.05$. The significance of the observed differences was assessed using one-way analysis of variance, and Bonferroni's *post-hoc* test was performed to determine which measurements were significantly different. *T*-tests were used to examine differences in anthropometrics and physical indicators between elite and sub-elite players.

## RESULTS

### Distribution of the ACE and ACTN3 in Elite and Sub-elite Players

The genotype distributions of controls ($\chi^2 = 2.793$, *df* $= 1$, $p = 0.09$; $\chi^2 = 3.678$, *df* $= 1$, $p = 0.055$, respectively), elite players ($\chi^2 = 0.085$, *df* $= 1$, $p = 0.77$; $\chi^2 = 3.567$, *df* $= 1$, $p = 0.059$, respectively), and sub-elite players ($\chi^2 = 1.7347$, *df* $= 1$, $p = 0.188$) were in agreement with the Hardy–Weinberg equilibrium, except for the *ACE* genotype ($\chi^2 = 5.652$, *df* $= 1$, $p = 0.017$) distribution of sub-elite players.

Table 4 summarize the distribution frequency of *ACTN3* and *ACE* genotypes and alleles in controls and elite and sub-elite players. There were significant difference in the distribution frequency of RR ($p = 0.024$) and DD ($p = 0.02$) between elite and sub-elite players. Furthermore, there were no significant differences in the distribution frequencies of *ACTN3* and *ACE* genotypes between controls, elite, and sub-elite players (Table 4). Additionally, there were significant differences in the frequencies of the R and X alleles of *ACTN3* between elite and sub-elite players ($\chi^2 = 6.898$, *df* $= 1$, $p = 0.009$). In addition, there were no significant differences in the frequencies of I and D alleles of *ACE* between controls and elite and sub-elite players.

Table 5 shows the association of *ACTN3* and *ACE* polymorphisms with controls and elite and sub-elite players. The OR of elite players in the RR *vs.* XX genotype compared

**Table 4** Genotype and allele frequencies of *ACTN3* R577X and *ACE* polymorphisms in elite and sub-elite players and controls.

|  | Control | Elite | Sub-elite | *p* |
|---|---|---|---|---|
| All (N) | 107 | 73 | 69 | |
| RR (N/%) | 26 (24.1%) | 27 (37.0%) | 14 (20.3%) | 0.054 |
| RX (N/%) | 43 (39.8%) | 28 (38.4%) | 28 (40.6%) | 0.943 |
| XX (N/%) | 38 (35.2%) | 18 (24.7%) | 27 (39.1%) | 0.207 |
| HWE-P value | 0.055 | 0.059 | 0.188 | |
| R allele (N/%) | 97 (46.3%) | 82 (56.2%) | 56 (40.6%) | 0.294 |
| X alele (N/%) | 119 (54.7%) | 64 (43.8%) | 82 (59.4%) | 0.318 |
| All (N) | 107 | 73 | 69 | |
| II (N/%) | 46 (43.0%0 | 31 (42.5%) | 28 (40.6%) | 0.969 |
| ID (N/%) | 42 (39.3%) | 34 (46.6%) | 24 (34.8%) | 0.396 |
| DD (N/%) | 19 (17.8%) | 8 (11.0%) | 17 (24.6%) | 0.066 |
| HWE-P value | 0.090 | 0.770 | 0.017 | |
| I allele (N/%) | 134 (62.6%) | 96 (65.8%) | 80 (58.0%) | 0.769 |
| D allele (N/%) | 80 (37.4%) | 50 (34.2%) | 58 (42.0% | 0.648 |

**Table 5** Odds ratios of *ACTN3* R577X and *ACE* genotypes for elite and sub-elite players.

|  | Elite *vs.* sub-elite | | Control *vs.* elite | | Control *vs.* sub-elite | |
|---|---|---|---|---|---|---|
|  | *p* | OR | *p* | OR | *p* | OR |
| RR *vs.* RX | 0.148 | 1.929 (0.840; 4.429) | 0.273 | 1.595 (0.777; 3.272) | 0.688 | 0.827 (0.370; 1.850) |
| RR *vs.* XX | 0.019 | 2.893 (1.201; 6.966) | 0.054 | 2.192 (1.007; 4.771) | 0.543 | 0.758 (0.335; 1.713) |
| RR *vs.* (RX + XX) | 0.041 | 2.306 (1.084; 4.906) | 0.070 | 1.829 (0.956; 3.499) | 0.584 | 0.793 (0.380; 1.653) |
| XX *vs.* (RR + RX) | 0.073 | 0.509 (0.248; 1.045) | 0.144 | 0.603 (0.311; 1.170) | 0.635 | 1.167 (0.625; 2.181) |
| II *vs.* ID | 0.578 | 0.782 (0.376; 1.623) | 0.625 | 0.832 (0.438; 1.582) | 0.863 | 1.065 (0.536; 2.118) |
| II *vs.* DD | 0.099 | 2.353 (0.880; 6.291) | 0.364 | 1.601 (0.623; 4.111) | 0.410 | 0.680 (0.304;1.522) |
| II *vs* (ID + DD) | 0.866 | 1.081 (0.554; 2.108) | 0.774 | 0.897 (0.51; 1.577) | 0.876 | 0.906 (0.490; 1.482) |
| DD *vs* (II + ID) | 0.046 | 0.376 (0.151; 0.941) | 0.288 | 0.570 (0.235; 1.383) | 0.339 | 1.514 (0.723; 3.169) |

**Notes.**
Data are odds ratio and 95% confidence intervals. Analysis was adjusted by competitive level.

with sub-elite players was 2.893 (95% CI [1.201–6.966]; $p = 0.019$). The OR of elite players in the RR *vs.* (RX + XX) genotype compared with sub-elite players was 2.306 (95% CI [1.084–4.906]; $p = 0.041$). The OR of elite players in the DD *vs.* (II + ID) genotype compared with sub-elite players was 0.376 (95% CI [0.151–0.941]; $p = 0.046$).

As shown in Table 6, both elite and sub-elite RR players' YYIR1 running distances were longer than those of RX players ($p = 0.05$ and $p = 0.025$, respectively). Elite XX players' VO$_2$ max was significantly higher than that of RX players ($p = 0.011$). Both elite and sub-elite players showed no significant difference in anthropometrics and athletic performance between II, ID, and DD players (Table 7). As shown in Table 8, elite players had greater VO$_2$max ($p = 0.024$). Moreover, their $5 \times 25$ m RSA ($p < 0.001$), 20-m ($p = 0.02$), and 30-m sprints ($p = 0.018$) times were shorter, and their standing long jumps ($p = 0.049$) and YYIR1 ($p = 0.037$) were longer than those of sub-elite players. The $5 \times 25$ m

**Table 6  Comparison of anthropometrics and athletic performance among different *ACTN3* genotypes.**

|  | Variables | ACTN3 | | | F | p |
|---|---|---|---|---|---|---|
|  |  | RR | RX | XX |  |  |
| Overall | Height (cm) | 172.9 ± 8.29 | 168.5 ± 8.74 | 169.6 ± 9.17 | 3.036 | 0.051 |
|  | Body mass (kg) | 56.16 ± 8.90 | 53.98 ± 9.22 | 54.24 ± 8.84 | 0.773 | 0.464 |
|  | C Thigh (cm) | 49.9 ± 3.91 | 50.3 ± 4.69 | 50.8 ± 5.63 | 0.367 | 0.693 |
|  | VO$_2$ max (ml/kg/min) | 54.96 ± 3.43 | 53.69 ± 5.05 | 55.34 ± 3.34 | 2.461 | 0.089 |
|  | 5 × 25m RSA (s) | 35.01 ± 1.91 | 35.88 ± 2.41 | 35.91 ± 1.52 | 2.778 | 0.066 |
|  | SLJ (cm) | 224.5 ± 18.2 | 216.4 ± 18.53 | 216.0 ± 18.60 | 2.927 | 0.057 |
|  | 20 m (s) | 3.34 ± 0.24 | 3.32 ± 0.29 | 3.37 ± 0.24 | 0.450 | 0.638 |
|  | 30 m (s) | 4.71 ± 0.44 | 4.68 ± 0.37 | 4.76 ± 0.32 | 0.554 | 0.576 |
|  | YYIR1 (m) | 1,913.2 ± 268.2[a] | 1,654.5 ± 361.7[a, b] | 1,811.1 ± 306.0[b] | 8.095 | <0.001 |
| Elite | Height (cm) | 173.3 ± 8.23 | 170.7 ± 8.13 | 173.1 ± 8.63 | 0.794 | 0.456 |
|  | Body mass (kg) | 57.31 ± 9.35 | 56.36 ± 8.84 | 58.68 ± 8.87 | 0.360 | 0.699 |
|  | C Thigh (cm) | 49.9 ± 4.06[c] | 51.8 ± 4.49 | 53.3 ± 3.34[c] | 4.026 | 0.022 |
|  | VO$_2$ max (ml/kg/min) | 55.24 ± 4.01 | 54.04 ± 4.46[b] | 57.68 ± 3.09[b] | 4.567 | 0.014 |
|  | 5 × 25m RSA (s) | 34.48 ± 1.74 | 34.87 ± 1.99 | 35.39 ± 1.48 | 1.392 | 0.255 |
|  | SLJ (cm) | 225.6 ± 19.14 | 221.4 ± 15.80 | 215.9 ± 17.11 | 1.679 | 0.194 |
|  | 20 m (s) | 3.31 ± 0.25 | 3.24 ± 0.24 | 3.34 ± 0.25 | 1.018 | 0.367 |
|  | 30 m (s) | 4.68 ± 0.41 | 4.57 ± 0.29 | 4.71 ± 0.34 | 1.051 | 0.355 |
|  | YYIR1 (m) | 1,942.2 ± 290.2[a] | 1,721.4 ± 362.3[a] | 1,853.3 ± 348.6 | 3.041 | 0.054 |
| Sub-elite | Height (cm) | 172.0 ± 8.62 | 166.3 ± 8.90 | 167.3 ± 8.93 | 2.000 | 0.144 |
|  | Body mass (kg) | 53.94 ± 7.79 | 51.60 ± 9.13 | 51.28 ± 7.61 | 0.514 | 0.600 |
|  | C Thigh (cm) | 49.9 ± 3.72 | 48.9 ± 4.50 | 49.0 ± 6.23 | 0.168 | 0.845 |
|  | VO$_2$ max (ml/kg/min) | 54.41 ± 1.89 | 53.35 ± 5.65 | 53.97 ± 2.64 | 0.355 | 0.703 |
|  | 5 × 25m RSA (s) | 36.03 ± 1.85 | 36.89 ± 2.40 | 36.25 ± 1.48 | 1.141 | 0.326 |
|  | SLJ (cm) | 222.29 ± 16.79 | 211.36 ± 20.07 | 216.07 ± 19.59 | 1.526 | 0.225 |
|  | 20 m (s) | 3.38 ± 0.24 | 3.41 ± 0.32 | 3.39 ± 0.24 | 0.040 | 0.960 |
|  | 30 m (s) | 4.78 ± 0.50 | 4.80 ± 0.41 | 4.80 ± 0.30 | 0.021 | 0.979 |
|  | YYIR1 (m) | 1,857.1 ± 218.5[a] | 1,587.5 ± 354.9[a] | 1,782.9 ± 277.3 | 4.724 | 0.012 |

**Notes.**

VO$_2$ max, maximum oxygen uptake; SLJ, standing long jump; 5× 25-m RSA, 5× 25-m repeated sprint ability; YYIR1, YoYo intermittent recovery test level 1.

[a]Significant difference between the RR and RX genotype: $p < 0.05$.

[b]Significant difference between the RX and XX genotype: $p < 0.05$.

[c]Significant difference between the RR and XX genotype: $p < 0.05$.

RSA time of RR ($p = 0.012$), RX ($p = 0.001$), II ($p = 0.013$) and ID ($p < 0.001$) elite players was significantly shorter than sub-elite players. Elite RX player's standing long jumps were longer ($p = 0.042$), and the 20-m ($p = 0.036$) and 30-m ($p = 0.022$) sprint times were significantly shorter than those of sub-elite players. Elite XX and II players were taller ($p = 0.036$ and $p = 0.026$) and heavier ($p = 0.05$ and $p = 0.001$) than sub-elite players. In addition, the VO$_2$ max of elite RX players was greater than that of sub-elite players.

## DISCUSSION

Our results showed that elite players were more likely to have the RR genotype compared with sub-elite players and controls. Furthermore, elite players were less likely to have the

**Table 7  Comparison of anthropometrics and athletic performance among different *ACE* genotypes.**

| Groups | | ACE | | | F | p |
|---|---|---|---|---|---|---|
| | Variables | II | ID | DD | | |
| Overall | Height (cm) | 170.7 ± 8.22 | 170.0 ± 9.94 | 169.1 ± 7.94 | 0.300 | 0.741 |
| | Body mass (kg) | 55.32 ± 8.04 | 54.20 ± 10.08 | 54.34 ± 8.72 | 0.246 | 0.783 |
| | C Thigh (cm) | 50.6 ± 4.35 | 50.2 ± 4.84 | 50.0 ± 5.74 | 0.205 | 0.815 |
| | VO₂ max (ml/kg/min) | 55.26 ± 3.14 | 54.09 ± 4.95 | 54.32 ± 4.28 | 1.232 | 0.295 |
| | 5 × 25 m RSA (s) | 35.52 ± 2.14 | 35.57 ± 2.11 | 36.06 ± 1.67 | 0.644 | 0.527 |
| | SLJ (cm) | 219.3 ± 20.01 | 218.8 ± 17.54 | 216.3 ± 18.65 | 0.230 | 0.795 |
| | 20 m (s) | 3.34 ± 0.28 | 3.33 ± 0.27 | 3.38 ± 0.20 | 0.350 | 0.705 |
| | 30 m (s) | 4.72 ± 0.40 | 4.69 ± 0.39 | 4.78 ± 027 | 0.467 | 0.628 |
| | YYIR1 (m) | 1,786.4 ± 332.0 | 1,797.4 ± 333.1 | 1,717.6 ± 354.0 | 0.518 | 0.597 |
| Elite | Height (cm) | 172.9 ± 7.96 | 171.8 ± 9.23 | 171.8 ± 5.15 | 0.164 | 0.849 |
| | Body mass (kg) | 58.46 ± 8.23 | 56.02 ± 9.68 | 58.06 ± 8.91 | 0.627 | 0.537 |
| | C Thigh (cm) | 51.17 ± 4.02 | 51.5 ± 4.44 | 52.4 ± 4.66 | 0.262 | 0.770 |
| | VO₂ max (ml/kg/min) | 55.85 ± 3.49 | 54.84 ± 4.93 | 55.86 ± 3.35 | 0.525 | 0.594 |
| | 5 × 25 m RSA (s) | 34.88 ± 2.28 | 34.67 ± 1.16 | 35.54 ± 1.94 | 0.755 | 0.474 |
| | SLJ (cm) | 223.3 ± 18.65 | 220.8 ± 15.61 | 218.4 ± 22.65 | 0.300 | 0.742 |
| | 20 m (s) | 3.30 ± 0.29 | 3.27 ± 0.20 | 3.37 ± 0.22 | 0.488 | 0.616 |
| | 30 m (s) | 4.64 ± 0.37 | 4.62 ± 0.35 | 4.76 ± 0.031 | 0.511 | 0.602 |
| | YYIR1 (m) | 1,852.3 ± 327.9 | 1,847.1 ± 321.4 | 1,722.5 ± 496.8 | 0.483 | 0.619 |
| Sub-elite | Height (cm) | 168.2 ± 7.91 | 167.5 ± 10.55 | 167.8 ± 8.81 | 0.039 | 0.962 |
| | Body mass (kg) | 51.84 ± 6.29 | 51.62 ± 10.27 | 52.59 ± 8.32 | 0.072 | 0.931 |
| | C Thigh (cm) | 50.0 ± 4.68 | 48.2 ± 4.83 | 48.8 ± 5.96 | 0.845 | 0.434 |
| | VO₂ max (ml/kg/min) | 54.6 ± 2.61 | 53.03 ± 4.88 | 53.60 ± 4.57 | 1.024 | 0.365 |
| | 5 × 25 m RSA (s) | 36.85 ± 2.48 | 36.30 ± 1.53 | 36.47 ± 1.98 | 0.682 | 0.509 |
| | SLJ (cm) | 214.9 ± 20.87 | 216.0 ± 19.97 | 215.4 ± 17.16 | 0.021 | 0.979 |
| | 20 m (s) | 3.38 ± 0.26 | 3.42 ± 0.33 | 3.39 ± 0.20 | 0.166 | 0.848 |
| | 30 m (s) | 4.80 ± 0.43 | 4.79 ± 0.43 | 4.79 ± 0.26 | 0.004 | 0.996 |
| | YYIR1 (m) | 1,713.6 ± 326.9 | 1,727.1 ± 343.5 | 1,715.3 ± 282.9 | 0.013 | 0.988 |

Notes.

VO₂ max, maximum oxygen uptake; SLJ, standing long jump; 5× 25-m RSA, 5× 25-m repeated sprint ability; YYIR1, YoYo intermittent recovery test level 1.

DD genotype than sub-elite players. However, the R and D alleles were not significantly different between elite and sub-elite players. Furthermore, the *ACE* and *ACTN3* genotypes were not associated with anthropometrics and athletic ability of Chinese football players, except that the RR genotype was associated with YYIR1.

In the meta-analysis, *McAuley et al. (2021b)* showed that *ACE* DD and *ACTN3* RR genotypes were associated with power-oriented phenotypes in football players. *Ma et al. (2013)* postulated that *ACE* II genotype was associated with endurance events, and that *ACTN3* R allele was associated with strength and power events. In addition, *Ahmetov & Fedotovskaya (2015)* suggested that *ACTN3* X and *ACE* I alleles were favorable for success in endurance athletes, and that the corresponding R and D alleles might be more favorable for athletes' performances in power/sprint.

**Table 8  Comparison of *ACE* and *ACTN3* genotypes athletic performances between elite and sub-elite football players.**

| | | | VO$_2$ max (ml/kg/min) | 5 × 25 m RSA (s) | SLJ (cm) | 20 m (s) | 30 m (s) | YYIR1 (m) |
|---|---|---|---|---|---|---|---|---|
| ACTN3 | RR | elite | 55.24 ± 4.01 | 34.48 ± 1.74 | 225.6 ± 19.14 | 3.31 ± 0.25 | 4.68 ± 0.41 | 1,942.2 ± 290.1 |
| | | sub-elite | 54.4 ± 1.88 | 36.03 ± 1.85 | 222.29 ± 16.79 | 3.38 ± 0.24 | 4.78 ± 0.50 | 1,857.1 ± 218.5 |
| | | *p* | 0.473 | 0.012 | 0.588 | 0.417 | 0.528 | 0.342 |
| | RX | elite | 54.04 ± 4.46 | 34.87 ± 1.99 | 221.3 ± 15.80 | 3.24 ± 0.24 | 4.57 ± 0.29 | 1,721.4 ± 362.3 |
| | | sub-elite | 53.35 ± 5.65 | 36.89 ± 2.40 | 211.4 ± 20.07 | 3.41 ± 0.32 | 4.80 ± 0.41 | 1,587.5 ± 354.9 |
| | | *p* | 0.615 | 0.001 | 0.042 | 0.036 | 0.022 | 0.168 |
| | XX | elite | 57.7 ± 3.09 | 35.39 ± 1.48 | 215.9 ± 17.11 | 3.34 ± 0.25 | 4.71 ± 0.34 | 1,853.3 ± 348.6 |
| | | sub-elite | 54.0 ± 2.64 | 36.25 ± 1.48 | 216.07 ± 19.59 | 3.39 ± 0.24 | 4.80 ± 0.30 | 1,783.0 ± 277.3 |
| | | *p* | <0.001 | 0.061 | 0.974 | 0.487 | 0.343 | 0.456 |
| ACE | II | elite | 55.85 ± 3.49 | 34.88 ± 2.28 | 223.3 ± 18.65 | 3.30 ± 0.29 | 4.64 ± 0.37 | 1852.3 ± 327.9 |
| | | sub-elite | 54.60 ± 2.61 | 36.85 ± 2.48 | 214.9 ± 20.87 | 3.38 ± 0.26 | 4.80 ± 0.43 | 1,713.6 ± 326.9 |
| | | *p* | 0.129 | 0.013 | 0.111 | 0.273 | 0.269 | 0.11 |
| | ID | elite | 54.84 ± 4.93 | 34.67 ± 1.16 | 220.8 ± 15.60 | 3.27 ± 0.20 | 4.62 ± 0.35 | 1,847.1 ± 321.4 |
| | | sub-elite | 53.03 ± 4.88 | 36.30 ± 1.53 | 216.0 ± 19.97 | 3.42 ± 0.33 | 4.79 ± 0.43 | 1,727.1 ± 343.5 |
| | | *p* | 0.173 | <0.001 | 0.311 | 0.039 | 0.097 | 0.179 |
| | DD | elite | 55.9 ± 3.35 | 35.54 ± 1.94 | 218.4 ± 22.65 | 3.37 ± 0.22 | 4.76 ± 0.31 | 1,722.5 ± 496.8 |
| | | sub-elite | 53.6 ± 4.57 | 36.47 ± 1.98 | 215.4 ± 17.16 | 3.39 ± 0.20 | 4.79 ± 0.26 | 1715.3 ± 282.9 |
| | | *p* | 0.226 | 0.181 | 0.714 | 0.801 | 0.826 | 0.963 |
| overall | | elite | 55.38 ± 4.19 | 34.85 ± 1.19 | 221.6 ± 17.58 | 3.29 ± 0.25 | 4.65 ± 0.35 | 1,835..6 ± 343.1 |
| | | sub-elite | 53.81 ± 4.02 | 36.47 ± 1.98 | 215.4 ± 19.42 | 3.40 ± 0.27 | 4.79 ± 0.39 | 1,718.7 ± 318.1 |
| | | *p* | 0.024 | <0.01 | 0.049 | 0.02 | 0.018 | 0.037 |

**Notes.**

VO$_2$ max, maximum oxygen uptake; SLJ, standing long jump; 5× 25-m RSA, 5× 25-m repeated sprint ability; YYIR1, YoYo intermittent recovery test level 1.

This study revealed that the *ACE* and *ACTN3* genotypes were not associated with muscle phenotype of Chinese youth football players. Moreover, studies on Italian and Polish athletes found no association between *ACE* and *ACTN3* and physical performance (*Massidda et al., 2012a*; *Massidda et al., 2012b*; *Myosotis et al., 2015*; *Orysiak et al., 2018*). These results suggest that the *ACE* and *ACTN3* genes may not be the major factor responsible for predisposing individuals to a particular sports event; it also affected by other factors, such as gene–gene interactions.

Studies have revealed that athletes with *ACE* DD genotype had greater muscle volume (*Charbonneau et al., 2008*) and strength (*Williams et al., 2005*), and were also associated with increased percentage of fast-twitch muscle fibers (*Zhang et al., 2003*). However, *Micheli et al. (2011)* reported that Italian *ACE* ID football players performed better in jumping than DD players. In contrast, this study showed that there was no relationship between the *ACE* I/D polymorphism and muscle phenotype. *Gentil et al. (2012)* suggested that *ACE* I/D polymorphisms were not associated with muscle strength in young men. Similarly, there was no association between muscle strength/power and ACE genotypes in untrained males (*Erskine et al., 2014*), nor in Indian army triathletes (*Shenoy et al., 2010*).

Previous studies have found that the *ACTN3* RR genotype or R allele was associated with muscle phenotype (*Shang et al., 2012*; *Erskine et al., 2014*; *Pimenta et al., 2013*). However, in this study, we did not find an association between the RR genotype and Chinese football players' muscle phenotypes. Furthermore, the *ACTN3* gene was not associated with muscle power in Spanish volleyball (*Ruiz et al., 2011*) and basketball (*Garatachea et al., 2014*) players.

In addition, in this study, we found that although the YYIR1 distance of RR players was significantly longer distance than that of RX players (Table 6), there was no significant difference in YYIR1 distance between elite and sub-elite RR players (Table 8). Above all, the effects of *ACE* and *ACTN3* genes on athletic performance in athletes are quite different (*Coelho et al., 2018*; *McAuley et al., 2021b*; *Santiago et al., 2008*; *Gineviciene et al., 2016*). We posit two reasons to explain these disparities.

The first and most important reason is the sample size. Insufficient sample size is currently the important reason for limiting the discovery of genotypes affecting physical performance (*Zilberman-Schapira, Chen & Gerstein, 2012*). *Hagberg et al. (2011)* suggested that at least 1,400 participants are needed to establish an association between one single nucleotide polymorphism and athletic performance. Insufficient sample size limits the accuracy of the results and reduces the value of significant difference (*Hagberg et al., 2011*). However, it is difficult to recruit a sufficient numbers of elite athletes in each sports event in a country or region.

The second reason is that the physical phenotype of the same gene can be affected by ethnicity, environment and other factors (*Zilberman-Schapira, Chen & Gerstein, 2012*; *Wang et al., 2013*). The *ACE* II genotype was associated with Russian athletes and the ID genotype for Lithuanian strength/power athletes. However, the ACE gene showed no association with muscle phenotypes in Polish athletes, although they are all Caucasian (*Gineviciene et al., 2016*; *Orysiak et al., 2018*). Moreover, the relation between *ACTN3* gene and athletic performance was also similar to that of *ACE* gene. Lithuanian *ACTN3* XX

genotype male athletes exhibited greater muscle power than RR athletes (*Ginevičiene et al., 2011*). In contrast to this study, Polish RR genotype athletes had significantly greater explosive power than that shown by XX athletes (*Orysiak et al., 2014*).

This study possesses several limitations. First, the sample size is small. The smaller sample size may affect the accuracy of the results, and the value of the significant differences may also decrease. Second, the elite players were from three different teams, and the practice environment, and coaching styles may have varied considerably between teams. These factors also play an important role when assessing the association between genotype and athletic performance. Finally, we recommend that future investigators employ longitudinal designs with large sample size to infer the effect of genotype on athletic performance.

## CONCLUSIONS

In conclusion, although there were significant differences in RR and DD genotype frequencies between elite and sub-elite players, the *ACE* and *ACTN3* genotypes were not associated with speed, leg explosive power, and repeated sprint ability in Chinese football players. Furthermore, the elite XX genotype of the *ACTN3* gene was associated with $VO_2$ max. The findings of this study may have future implications for talent identification for Chinese youth football teams. Strength and conditioning coaches may adjust the training load by taking into account a player's genetic information.

### Funding
This work was supported by the General Administration of Sport of China (No. 2022-B-04). The funders had no role in study design, data collection and analysis, decision to publish, or preparation of the manuscript.

### Grant Disclosures
The following grant information was disclosed by the authors:
General Administration of Sport of China: 2022-B-04.

### Competing Interests
The authors declare there are no competing interests.

### Author Contributions
- Shidong Yang conceived and designed the experiments, performed the experiments, analyzed the data, prepared figures and/or tables, authored or reviewed drafts of the article, and approved the final draft.
- Wentao Lin conceived and designed the experiments, analyzed the data, prepared figures and/or tables, authored or reviewed drafts of the article, and approved the final draft.
- Mengmeng Jia conceived and designed the experiments, analyzed the data, prepared figures and/or tables, authored or reviewed drafts of the article, and approved the final draft.

- Haichun Chen conceived and designed the experiments, authored or reviewed drafts of the article, and approved the final draft.

## Human Ethics

The following information was supplied relating to ethical approvals (i.e., approving body and any reference numbers):

The Ethics Committee of Fujian Normal University approved the study.

## Data Availability

The raw measurements are available as a Supplementary File.

## Supplemental Information

Supplemental information for this article can be found online at http://dx.doi.org/10.7717/peerj.14893#supplemental-information.

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
