# Peer review of "Association between *ACE* and *ACTN3* genes polymorphisms and athletic performance in elite and sub-elite Chinese youth male football players"

_PeerJ, doi:10.7717/peerj.14893_

## Round 0.1 · original submission · Major Revisions

Dear authors. Please note all recommendations about your manuscript. I gently did and attached a file with the comments of all reviewers hoping to facilitate your revision.

Reviewer 1 ·

Basic reporting

Thank you for the opportunity to review this paper titled “ACE and ACTN3 genes polymorphisms and Athletic performance in elite and sub-elite Chinese youth male football players”. The paper is clearly written, and the topic of genes polymorphisms of football is something that’s likely to be of interest to many working in the area of sports. I have, however, provided a number of comments for each section for the authors to consider.

General comment
Can you explain what is an elite football player for you? How were elite football players defined?
Grammar corrections throughout the paper are needed.

Abstract:
Could you provide specifically the age of the different groups of the players?
Change “used to compare” by “used to observe the association” or”relationship”
Please correct the spaces between “=”
Could add some practical indications in conclusion section?

Introduction
About following statements“Ma and his coworker suggested that the R allele was only associated with power athletes by meta-analysis (Ma et al., 2013a)” I would recommend removing this sentence and please, add information about other articles that found no relationship between genotype and performance in football. The authors should review the article https://pubmed.ncbi.nlm.nih.gov/25650734/
Line 60-65: This point is made very quickly. Can you provide a bit more detail for your reader.
Line 88-90: This study does not hold the potential to answer this hypothesis. I recommend to delete it.
I wonder whether alpha-actinin-3 deficiency is football specific or not.

Methods section:
Line 93-96: General comment regarding participants: What happened if some football player had any lower limb pain (very usual in football) before measurement? Was he excluded? And if some football player suffers any tackle in the match? You should clarify these questions in exclusion criteria, because you don´t show if any player dropped out from the study. How many people were initially recruited and how many failed to qualify due to not meeting the inclusion/exclusion criteria? Need more detail within this section.
Line 95: wrong word berfoe.
Line 97-99: ethics approval number should be provided. Additionally, it should be noted if the protocol was pre-registered.
Line 110-128: please cite validation work for this gene-expression assay system.
Line 133-172: Was any warm-up completed prior to the measurements?
Line 158: what is the reliability of the aerobic fitness measurements? How the participants familiarized with the measurements?.
Goalkeepers are included in the study?. In my opinion, the goalkeepers should be excluded from the study because of the different nature of their activity.

Results
Figure 1 and Figure 2: It would be easier for the reader to identify distributions if the percentage values were also placed inside each category of the stacked bar.
Line 195: wrong word Talbe.
Line 195: order the numbering of the tables, it cannot appear table 2 and 3 after table 4, 5, 6 and 7 in the text.
The results section is a little bit too long and wordy with some text parts more or less duplicated.

Discussion
Is there evidence that ethnicity has effects on the expression of genes investigated in this study? Were all recruited subjects China?
Line 209-213: You should remove the first and second sentence of the discusión section. The first paragraph should state the main findings.
Line 220-227: the included studies used comparable methodologies?
Line 237-247: the paragraph is currently very wordy, can you try and make the message more succinct.
Lack of information regarding what training included on field. As there are 3 separate schools (two) and profession football club in China (one) involved a lot more information is required as to the training and match schedules of the clubs and the individual players. For example, were some of the players predominantly substitutes or starting players?
Conclusions
Could add some practical indications.

I have two major comment for your study.
1) Sample size is very small. Thus, the potential for a type II error is high. Especially you discuss about complex of football performance in DISCUSSION, you need more sample size to conclude the association of the genotype and football performance.
2) Control subjects. You need control subjects and more statistical analysis. See study of Eynon et al. in European populations. (PMID 22916217, 23522773).

Experimental design

Methods described with no sufficient detail.

Validity of the findings

Interesting and original article.

·

Basic reporting

Please go to additional comments.

Experimental design

Please go to additional comments.

Validity of the findings

Please go to additional comments.

Additional comments

The manuscript is interesting. New in terms of publishing contrasting results and not attached to the hypothesis suggested by the literature. However, for it to be published, it is necessary to improve the complete presentation of the manuscript.
Firstly, improve the writing and the correct writing of English.
The title should mention that the work is an association study.

Introduction: The authors should clarify the problem and the question before presenting the work's purpose and the research hypothesis.
Although it is an association study, the authors must present the known biochemical, physiological, and molecular arguments behind said associations and discuss their results based on them. Without this request, the manuscript becomes a simple association and descriptive study of little interest. In this sense, their references are appropriate but little used. There are several possible mechanisms by which ACE and ACTN3, AND NOT OTHER GENES, exert their functions IN SKELETAL MUSCLE and connective tissue, functions that favor specific physical capacities. Please check your references better and/or consult others, for example:
Taylor, R.R., Mamotte, C.D., Fallon, K., & van Bockxmeer, F.M. (1999). Elite athletes and the gene for angiotensin-converting enzyme. Journal of Applied Physiology, 87(3), 1035-1037.
Kostka, J., Sikora, J., & Kostka, T. (2017). Relationship of quadriceps muscle power and optimal shortening velocity with angiotensin-converting enzyme activity in older women. Clinical interventions in aging, 12, 1753.
Onder, G., Penninx, B. W., Balkrishnan, R., Fried, L. P., Chaves, P. H., Williamson, J., ... & Pahor, M. (2002). An observational study is a relationship between the use of angiotensin-converting enzyme inhibitors and muscle strength and physical function in older women. The Lancet, 359(9310), 926-930.
Or the Wikipedia articles.
Please improve the presentation and information in Tables 5, 6, and 7. They should understand by themselves without having to go to the text. Put the units of each variable. At the foot of the table, add the information on the acronyms. Table 7 is impossible to read.
In the text of the results, say which figure or table it refers to.
Please remove the penultimate paragraph (lines 269-279); it has nothing to do with this work.
The conclusion is the conclusion, not a discussion.

·

Basic reporting

Thank you for the opportunity to review the research entitled "ACE and ACTN3 genes polymorphisms and athletic performance in elite and sub-elite Chinese youth male football players."

There is great enthusiasm about the future relevance of genomics in the sports/exercise field. During the past two decades, tools to understand how genetic underpinnings such as single nucleotide polymorphisms (SNPs) influence exercise performance phenotypes have evolved at a rapid rate, resulting in a substantial number of publications. However, only a few meet an acceptable standard as defined by sample size, measurement quality, study design, and genotyping quality [Overcoming Barriers to Progress in Exercise Genomics. Exerc. Sports Sci. Rev. 39(4):212-217].

Although the authors recognize the study limitations due to the small sample size, I believe that it would be better described as a pilot study. Furthermore, due to the lower statistical power to detect an effect when one exists, it may be better to just describe the genotypic distribution between groups.

Experimental design

The authors describe that a sample of elite and sub-elite athletes was evaluated. However, the method section is unclear how this specific sample is characterized. Is this a convenience sampling? How was the subject's training routine considered "elite"? How can school players be characterized as 'sub-elite'?

The Methods section is too superficial and needs to be better described and detailed. A brief (at least) description of the key measurements is required for the reader to interpret the findings' significance fully. What quality controls were performed for the SNPs analyses?

Comparing anthropometric measurements in adolescents can be tricky and affected by confounders (age 13-15 years). Did the authors measure the sexual maturation of the subjects?

It is possible to estimate VO2max using the Yo-Yo Intermittent Recovery Level 1 test (specific to soccer). Why did the authors use a continuous 12-minute running test (non-soccer specific) to estimate VO2max?

Validity of the findings

Statistical analysis must be reviewed. The chi-square test on the observed and expected values is performed to see if the observational data support the hypothesis that the population is at Hardy-Weinberg equilibrium for the gene. In Hardy-Weinberg chi-square analysis, the number of degrees of freedom is equal to the number of genotypes minus the number of alleles. In this case, 3 – 2 = 1. Therefore, it uses the chi-square distribution with 1 degree of freedom.

As an example, my analysis below got a different result.
ACE indel polymorphism - Elite (II = 31; ID = 34; DD = 8): x2 = 0.085; *p-value = 0.770
ACE indel polymorphism - Sub-Elite (II = 28; ID = 24; DD = 17): x2 = 5.652; *p-value = 0.017

* With 1 degree of freedom.

---

## Round 0.2 · Minor Revisions

After corrections, we think that the manuscript improved significantly. However, after a second revision, again the legitimate concern about the lack of a control group and the size of your "n", which can greatly increase the size of the type II error. Please, see the revisions.

Reviewer 1 ·

Basic reporting

The article is well designed and interesting. However, Sample size is very small and they have no control subjects. There are some specific comments, which are detailed below.

Minor comments:
Method of genotyping is not clearly described. How do you verify your results, sequencing or something else? Please, specify and provide sources. ACTN3 genotype frequency could vary depending on studied population. All tested persons were Chinese or not? Also, have you compared athletes to control ones?
Have you checked the diet of players?

Clearly state the inclusion criteria. How was the recruitment conducted?

Please, name the ethics committee and give the registration number.

This statement “The distance from the line to the player’s closest heel was then measured with a measuring tape.“ needs a reference.

Please correct the spaces between “=” in the document.

Please be consistent, minutes or min?

Experimental design

Is ok

Validity of the findings

Major comments:
I have two major comments for your study.
1. Sample size
Sample size is very small. Thus, the potential for a type II error is high. Especially you discuss about complex of football performance in DISCUSSION, if effect size of the genetic factors is small, you need more sample size to conclude the association of the genotype and Chinese youth male football performance.

2. Control subjects.
You need control subjects and more statistical analysis. See study of Eynon et al. in European populations. (PMID 22916217, 23522773).

Additional comments

The article is well designed and interesting. However, Sample size is very small and they have no control subjects. There are some specific comments, which are detailed below.

Minor comments:
Method of genotyping is not clearly described. How do you verify your results, sequencing or something else? Please, specify and provide sources. ACTN3 genotype frequency could vary depending on studied population. All tested persons were Chinese or not? Also, have you compared athletes to control ones?
Have you checked the diet of players?

Clearly state the inclusion criteria. How was the recruitment conducted?

Please, name the ethics committee and give the registration number.

This statement “The distance from the line to the player’s closest heel was then measured with a measuring tape.“ needs a reference.

Please correct the spaces between “=” in the document.

Please be consistent, minutes or min?

Major comments:

I have two major comments for your study.

1. Sample size
Sample size is very small. Thus, the potential for a type II error is high. Especially you discuss about complex of football performance in DISCUSSION, if effect size of the genetic factors is small, you need more sample size to conclude the association of the genotype and Chinese youth male football performance.

2. Control subjects.
You need control subjects and more statistical analysis. See study of Eynon et al. in European populations. (PMID 22916217, 23522773).

·

Basic reporting

The document has been substantially improved in all its parts, but with a few grammatical details.

Experimental design

no comment

Validity of the findings

No comment

Additional comments

Good job. Congratulations

---

## Round 0.3 · accepted · Accept

Congrats, your article was accepted. After the correction of all questions about your manuscript, you got an impressive evolution. I hope that you understand that our evaluation process is rigorous and impartial, which can be a hard task, however, this results in high-quality manuscripts.

Please, pay attention to the next steps of our publication process. Peer J will contact you during the process.

Regards.